# Bioactivity Evaluation of a Novel Formulated Curcumin

**DOI:** 10.3390/nu11122982

**Published:** 2019-12-06

**Authors:** Se-Chun Liao, Wei-Hsiang Hsu, Zi-Yi Huang, Kun-Lin Chuang, Kuan-Ting Lin, Chia-Ling Tseng, Tung-Hu Tsai, Anh-Hoang Dao, Chun-Li Su, Chi-Ying F. Huang

**Affiliations:** 1Institute of Clinical Medicine, National Yang-Ming University, Taipei 112, Taiwan; susanliao@everestpharm.com; 2Institute of Biopharmaceutical Sciences, National Yang-Ming University, Taipei 112, Taiwan; rabbitjim5@hotmail.com (W.-H.H.); jason101024004@gmail.com (K.-L.C.); 3Cold Spring Harbor Laboratory, Cold Spring Harbor, NY 11724, USA; woodydon777@gmail.com; 4Department of Human Development and Family Studies, National Taiwan Normal University, Taipei 106, Taiwan; piiq2005@hotmail.com; 5Institute of Traditional Medicine, School of Medicine, National Yang-Ming University, Taipei 112, Taiwan; thtsai@ym.edu.tw; 6Department of Formulation and Processing, National Institute of Medicinal Materials, Hanoi 100000, Vietnam; 7Graduate Program of Nutrition Science, School of Life Science, National Taiwan Normal University, Taipei 106, Taiwan; 8Department of Biochemistry, College of Medicine, Kaohsiung Medical University, Kaohsiung 807, Taiwan

**Keywords:** curcumin, formulated curcumin, pharmacokinetics, aurora kinase A, hepatocellular carcinoma

## Abstract

Curcumin has been used as a traditional medicine and/or functional food in several cultures because of its health benefits including anticancer properties. However, poor oral bioavailability of curcumin has limited its oral usage as a food supplement and medical food. Here we formulated curcumin pellets using a solid dispersion technique. The pellets had the advantages of reduced particle size, improved water solubility, and particle porosity. This pellet form led to an improvement in curcumin’s oral bioavailability. Additionally, we used the C-Map and Library of Integrated Network-Based Cellular Signatures (LINCS) Unified Environment (CLUE) gene expression database to determine the potential biological functions of formulated curcumin. The results indicated that, similar to conventional curcumin, the formulated curcumin acted as an NF-κB pathway inhibitor. Moreover, ConsensusPathDB database analysis was used to predict possible targets and it revealed that both forms of curcumin exhibit similar biological functions, including apoptosis. Biochemical characterization revealed that both the forms indeed induced apoptosis of hepatocellular carcinoma (HCC) cell lines. We concluded that the formulated curcumin increases the oral bioavailability in animals, and, as expected, retains characteristics similar to conventional curcumin at the cellular level. Our screening platform using big data not only confirms that both the forms of curcumin have similar mechanisms but also predicts the novel mechanism of the formulated curcumin.

## 1. Introduction

The use of nutraceutical or functional and medical foods as alternative medicine, in addition to supplementary foods, has been on the rise in recent years [1,2]. The delivery of active ingredients is important in order to obtain beneficial effects for the human body.

Curcumin has been used for many years as a naturally occurring alternative medicine and functional food for the treatment of many diseases. Curcumin is a polyphenol extracted from the rhizome of *Curcuma longa* L., which has phenolic groups and conjugated double bonds [3]. It has strong anti-oxidant, anti-inflammatory, anti-septic, anti-proliferative, and wound-healing properties [4,5,6,7,8]. In addition, curcumin can reverse multidrug resistance of cancer cells, suggesting that it can also serve as a supplement to traditional chemotherapy [9,10]. Several lines of evidence show that curcumin exerts potent anticancer effects against a broad range of human cancer cells, including prostate, colon, breast, ovarian, lung, and liver cancers, and can induce cancer cell apoptosis, for example, in liver cancer cell lines, including but not limited to HepG2, SK-Hep-1, Hep3B, SUN449, and Huh7 cells, with low cytotoxic effects on normal cells [11,12,13,14,15,16].

Despite curcumin’s beneficial effects, its low oral bioavailability (due to its low absorption in the gut because of low solubility in water, fast metabolism by the liver, and rapid systemic elimination) has limited its applications [17,18,19]. Various methods have been developed to improve the oral bioavailability of curcumin, such as the use of a natural enhancer, a curcumin-phospholipid complex, cyclodextrin and microemulsions, and the development of curcumin analogs [20,21,22]. In this study, we selected the pellet form, a multiple-unit dosage form, as a vehicle for compound delivery. Pellets disperse freely in the gastrointestinal tract, so they invariably maximize drug absorption [23]. In addition, to improve the solubility of curcumin, we used a solid dispersion technique to formulate pellets. In solid dispersions, the particle size of poorly soluble drugs is reduced and their wettability and dispersibility enhanced, thereby improving their dissolution and absorption rate [24].

The Connectivity Map (C-Map) is a systematic database that establishes the relationship between diseases, genes, and compounds [25]. Recently, the database has been expanded and renamed C-Map and Library of Integrated Network-Based Cellular Signatures (LINCS) Unified Environment (CLUE) (https://clue.io/) [26]. Briefly, C-Map and CLUE use gene expression profiles to describe the biological states of cultured human cancer or normal cells to determine their chemical or genetic constructs (short hairpin RNA [shRNA] constructs). The new, low-cost, high-throughput generic solution for gene expression profiles is termed “L1000.” CLUE not only expands the 1309 compounds listed in C-Map to 19,811 small molecules but also includes 5075 shRNA and overexpression genes. In addition, users find CLUE relatively convenient to quickly search for a drug class with a similar mechanism of action (MOA) as a target drug or the same gene family of genetic perturbagens, they codify the class-level annotation required considerable effort, perturbagen classes (PCLs). For example, users can compare their target, such as a disease gene signature or a novel compound, with C-Map and CLUE through pattern-matching algorithms and predict dissimilarities (search for a drug to reverse a disease) or similarities (search for a similar MOA via known compounds). They could upload gene expression profiles to C-Map and CLUE to calculate the connectivity score of each profile. A positive connectivity score would indicate a degree of similar mechanism, while a negative connectivity score would denote the reverse.

HCC is the fifth-most common malignancy worldwide. More than 75% of HCC cases occur in the Asia-Pacific region. The high mortality rate because of HCC is due to the difficulty in diagnosis and poor prognosis. Chemotherapy is a traditional choice for inoperable HCC, but drug resistance limits the therapeutic effect [27,28]. Sorafenib is a multi-kinase inhibitor that targets Raf kinases as well as vascular endothelial growth factor receptor (VEGFR)-2/VEGFR-3, platelet-derived growth factor receptor beta (PDGFR-β), Flt-3, and c-Kit. Because of its potential in providing a survival advantage of two to three months, as per results of two-phase III clinical studies, sorafenib is a Food and Drug Administration (FDA)-approved, first-line targeted therapy agent for treating advanced HCC patients [29,30]. However, the low tumor response rate and side effects of sorafenib indicate the need for investigating other new potential drugs or supplementary foods for HCC [31,32]. In this study, we investigated the anticancer activity of conventional and formulated curcumin and their combination with sorafenib in order to determine whether this combination can induce HCC cell apoptosis and autophagy and inhibit HCC cell proliferation. Formulated curcumin can be used as a functional food and alternative medicine in cancer therapy as it not only causes mitotic defects and cell cycle arrest in cancer cells but also alters chemosensitivity toward anticancer drugs by inducing Aurora-A suppression.

## 2. Materials and Methods

### 2.1. Materials and Methods Used in Manufacturing Formulated Curcumin

#### 2.1.1. Preparation of Curcumin Solid Dispersion Loaded Pellet

A curcumin standard with a purity > 95.6% was purchased from Sigma-Aldrich.

We used a solid dispersion technique to enhance the solubility and dissolution rate of curcumin [24]. Briefly, the process of making formulated curcumin included dispersing curcumin powder into a solid dispersion solution and spraying it onto sugar spheres. Tumeric extract powder contained 95% curcumin in 80 g (as the active drug). The excipients used in the preparation process of solid dispersion curcumin were Polyvinylpyrrolidone #k30 800 g (PVP K30, as the non-volatile polymer solvent for curcumin) and alcohol 3200 g (as the volatile solvent for curcumin). The turmeric extract powder was mixed with the excipients. The drug-polymer interaction evenly dispersed curcumin in the solvent. Next, the solvent containing solid dispersion curcumin was loaded onto sugar spheres by spray-drying to make solid dispersion pellets; the solvent evaporated during fluid-bed granulation (Figure 1).

#### 2.1.2. Measurements of Particle Size and Zeta Potential

Particle size (Z-average, nm), polydispersity index (PDI), and ζ-potential (ZP, mV) of curcumin particles after re-dispersion in water were determined at 25 °C by dynamic light scattering (DLS) using a Zetasizer Nano ZS90 (Malvern Instruments, Malvern, UK). Measurements were performed with a detector at a fixed angle of 90°, in triplicate, and results were shown as mean ± SD. About 60 mg of pellets were dispersed into 2 mL water and centrifuged at 2000 × g for 1 minute to remove starch particles originated from sugar spheres. Supernatant was aspirated to measure size and ZP.

#### 2.1.3. Dissolution Test

A dissolution test was conducted by the US Pharmacopeia 41 basket method (apparatus 1) using a dissolution tester (708-DS Dissolution apparatus, Agilent, USA). The samples of 89 mg conventional curcumin (equivalent to 72 mg curcumin) and 2180 mg formulated curcumin (equivalent to 72 mg curcumin) were placed into 900 mL of dissolution medium containing 1% sodium dodecyl sulfate (SDS) at 37 °C ± 0.5 °C, under a stirring speed of 100 ± 2 rpm. A 5 mL sample was withdrawn at each time interval (5, 10, 15, 20, 30, 45, and 60 minutes) and was mixed and filtered through a 0.45-µm pore membrane. Then, 2 mL of filtrate was diluted with mobile phase so that the total volume became 10 mL and subjected to HPLC analysis (Agilent 1260, USA). The HPLC program consisted of a mobile phase of tetrahydrofuran: 0.1% citric acid solution (4:6). The column used was 4.6 mm × 20 cm with 5-µm packing L1. The flow rate was 1 mL/minute, and the injection volume was 20 µL using a 420 nm wavelength detector. The percentages of curcumin dissolved from the conventional curcumin and pellets (formulated curcumin) into the medium were calculated and compared.

### 2.2. Pharmacokinetic Study

#### 2.2.1. Animal Model

All animal treatment procedures followed the Guide for the Care and Use of Laboratory Animals (National Institutes of Health (NIH) publication, 85–23, revised 1996) as well as the Animal research: Reporting *in vivo* experiments (ARRIVE) guidelines, and were approved by the Animal Research Committee at National Yang-Ming University, Taipei, Taiwan, under Institutional Animal Care and Use Committees (IACUC) approval no: 990103. All surgeries and experimental procedure were carried out under anesthesia with all efforts to minimize animal suffering.

Twelve male Sprague Dawley (SD) rats (270 ± 15 g body weight) were obtained from Bio-Lasco, Taipei, Taiwan. Water was provided *ad libitum*, regardless of administration route. All animals were acclimatized and quarantined in quarantine room of the Rosetta animal facility for about 1 week, and then transferred to feeding room. The humidity and temperature were well controlled as 30%–70% and 19–25 °C. The light and dark cycle was set as 12 h: 12 h. Food and drinking water were allowed *ad libitum* during housing.

Rats were randomly divided into two groups treated with curcumin and formulated curcumin. Curcumin (conventional; 500 mg/kg, *n* = 6) and 500 mg/kg formulated curcumin (equal to curcumin 60 mg/kg, *n* = 6) were administrated by gavage to the freely moving rats, respectively. A 300 μL blood sample was collected from the tail vein into a tube rinsed with heparin at 0, 0.25, 0.5, 1, 1.5, 2, 4, 6 h after oral administration.

#### 2.2.2. Sample Pretreatment

Plasma was obtained by centrifuging the blood sample at 4000 rpm for 10 minutes at 4 °C. The 10 μL plasma was mixed with 50 μL of the internal standard solution containing 0.1 ng/µL of agomelatine. The samples were vortexed and centrifuged at 13,000 rpm for 5 minutes. The 50 µL of supernatant was transferred to the 1.5 mL tube contained 50 µL solution of 25% acetonitrile and 0.1% acetic acid. After that, 50 μL of the solution was injected onto LC-MS system.

#### 2.2.3. LC/MS/MS Conditions and Data Analysis

Curcumin concentrations in the samples were determined by positive ion electrospray tandem mass spectrometry using multiple reaction monitoring (MRM). Separation of curcumin was conducted on a Cosmosil column (5C18-MS-IIPacked column, 120 Å, 5 µm, 4.6 mm I.D. x 150 mm; NACALAI TESQUE, Inc., Japan) with a mobile phase of acetonitrile-water-formic acid. MS/MS conditions consisted of a declustering potential of 50 V, desolvation temperature of 550 °C, spray needle of 5500 V, and collision energy of 30 V.

Pharmacokinetic analysis was calculated using a non-compartmental model with the Phoenix WinNonlin® (Version 8.0) software. The area under the drug concentration-time curve (AUC) was used to measure the total amount of curcumin reaching the systemic circulation. The relative oral bioavailability (BA) of curcumin was calculated according to the following equation: BA (%) = 100 × [(AUC_formulated curcumin_/dose_formulated curcumin_]/[(AUC_curcumin_/dose_curcumin_)]. The pharmacokinetic results were represented as the mean ± SD. Statistical analysis was performed by *t* test (SPSS version 10.0) to compare the differences between groups. The level of significance was set at *p* < 0.05.

### 2.3. Cell Lines and Cell Culture

The Huh7 and PLC5 cell lines were obtained from National Taiwan University Hospital, Taiwan. The Mahlavu cell lines were provided by Dr. Muh-Hwa Yang (Institute of Clinical Medicine, National Yang-Ming University, Taiwan). Hep3B cells were obtained from American Type Culture Collection (ATCC), Rockville, MD, USA.

HCC cell lines were cultured in Dulbecco’s modified Eagle’s medium (DMEM, GIBCO) supplemented with 10% (*v*/*v*) fetal bovine serum (FBS, GIBCO), non-essential amino acids (NEAA, GIBCO), L-glutamine (GlutaMAX™-I Supplement, GIBCO) and 10% penicillin-streptomycin (GIBCO). These cells were maintained in a humidified incubator with 5% CO_2_ at 37 °C and were regularly subcultured every 2–3 days.

### 2.4. Drug Preparation and Cell Exposure

The conventional curcumin was prepared as a 30 mM stock solution in dimethyl sulfoxide (DMSO; Sigma) and stored at -20 °C. Final curcumin concentrations of 1–90 μM were obtained by dilution in culture medium so that the final concentration of DMSO was less than 1%. Controls contained 0.1% DMSO in all experiments.

The formulated curcumin was prepared as a 30 mM stock solution (based on the weight of curcumin) in double distilled water and stored at -20 °C. The formulated curcumin was diluted with cell culture medium to obtain the concentration indicated.

### 2.5. Proliferation and Viability Assays

Cells were seeded into a 96-well plate (1500–2000 cells/well) overnight and then treated with curcumin and the formulated curcumin respectively for 0, 24, 48, 72, 96 and 120 h. After treatment, 0.5 μg/mL 3-(4,5-cimethylthiazol-2-yl)-2,5-diphenyl tetrazolium bromide (MTT) was added to each well and cultured for 2 h at 37 °C. After incubation, the media were removed from the wells. The formazan crystals formed were then solubilized in DMSO at room temperature for 10 minutes, and then the absorbance was measured in a multimode microplate reader at 570 nm.

### 2.6. Mitochondrial Membrane Potential Assay

We employed 5, 5’, 6, 6’-tetrachloro-l, 1’, 3, 3’-tetraethylbenzimidazolcarbocyanine iodide (JC-1), which was obtained from Cayman Chemical Co., to analyze the mitochondrial membrane potential. The cells were seeded in 96-well black plates at a density of 7000 cells/well and cultured overnight. After treatment, JC-1 staining solution was added to each well and incubated at 37 °C for 15–30 minutes in the dark. The plates were obtained by centrifuged at 400 × *g* at room temperature for 5 minutes, and the supernatant was discarded. Then, JC-1 assay buffer was added to each well, followed by centrifugation at 400 × *g* at room temperature for 5 minutes, after which the supernatant was discarded. Finally, JC-1 assay buffer was added again to each well for fluorescent analysis using a fluorescent plate reader.

### 2.7. Annexin V and Propidium Iodide (PI) Double Staining by Flow Cytometry

The Huh7 were incubated with various concentrations of conventional and formulated curcumin for 24 h. Annexin V/PI staining was performed to quantify cell apoptosis using an Annexin V-fluorescein isothiocyanate (FITC) Apoptosis Detection Kit (BioVision, Inc., Milpitas, CA, USA) according to the manufacturer’s protocol. Annexin V-FITC was then added followed by incubation for 15 minutes in the dark in a 100 µL cell suspension. PI was then spiked into 400 µL Annexin V binding buffer and added immediately to the cell suspension, and subsequently analyzed on a FACScan flow cytometer (BD Biosciences, USA).

### 2.8. Western Blot

The cells were incubated with various treatment (conventional and formulated curcumin or combination of sorafenib and conventional and formulated curcumin), and then collected for western blot. Aliquots of cell lysates containing 20–50 μg of protein were separated by SDS-polyacrylamide gel electrophoresis (SDS-PAGE), transferred onto a polyvinylidene difluoride (PVDF) membrane and detected using specific primary and secondary antibodies. The protein bands were visualized by an enhanced chemiluminescence (ECL) detection kit (Immobilon^TM^ western, Millipore). The membranes were reprobed for β-actin as a loading control. All western blots were carried out at least three times for each experiment. The data were normalized to β-actin. The following primary antibodies were used: anti-extracellular regulated protein kinases (ERK), anti-caspase-3, anti-poly (ADP-ribose) polymerase (PARP) (all from Cell Signaling Technology) and anti-aurora kinase A (AURKA) (BD Biosciences). All antibodies were used at a 1:1000 dilution.

### 2.9. Cell Cycle Analysis

After treatment, the cells were collected by trypsinization and fixed in precooled 70% ethanol overnight. The cells were then incubated with PI in the presence of RNase A. The DNA content was analyzed by a FACSCalibur, and the data were analyzed by Flowjo software. The percentage of cells in the sub-G1 was used to indicate the apoptosis rate.

### 2.10. Analysis the Similar Mechanism of Gene Expression Profiles of Conventional and Formulated Curcumin Using the L1000 Microarray

Human HCC (HepG2) and human colorectal cancer (HT29) cell lines (ATCC) were treated in triplicate with 20 μM of conventional curcumin or 2 μM of formulated curcumin. Briefly, for experiments using formulated curcumin, 2 μM of curcumin formulation and curcumin excipient were dissolved in DMSO and incubated with HepG2 and HT29 cell lines. The samples were submitted to Genometry, Inc. (Cambridge, MA, USA) for L1000 microarray analysis and to obtain the gene expression profiles of formulated curcumin in HepG2 and HT29. Each set of gene expression profiles consisted of up- and down-regulated gene signatures. Subsequently, CLUE was used to decipher the gene signatures in order to uncover potential mechanisms via mapping to compounds with known MOAs. To filter output data, we used a score > 90 for compounds and a score > 70 for PCLs.

### 2.11. Statistical Analysis

All values were expressed as the mean ± SD. The data were analyzed using a two-tailed student’s *t* test. A *p* < 0.05 was considered as statistically significant.

## 3. Result

### 3.1. Preparation and Evaluation Formulated Curcumin

First, we prepared the formulated curcumin from powder to pellets with a solid dispersion technique as described in detail in the Materials and Methods section. The pellet size of formulated curcumin estimated by sieving was distributed in the range of 830–1000 μm. Curcumin content quantified by HPLC was 3.3%. Z-average (nm), PDI, and ZP (mV) of curcumin particles after re-dispersion in water were 141.9 ± 5.1, 0.308 ± 0.029 and −2.49 ± 0.39, respectively (Table 1). The dissolution rates of conventional curcumin were 0.00%, 9.80%, 13.60%, 14.05%, 17.28%, 19.41%, 24.06%, and 26.28%, while those of the formulated curcumin were 0.00%, 59.03%, 87.50%, 98.78%, 100.22%, 101.20%, 102.29%, and 103.65% at 0, 5, 10, 15, 20, 30, 45, and 60 minutes, respectively (Appendix A). In the 1% SDS medium, more than 85% of curcumin was almost immediately released from the pellets after 10 minutes. Additionally, when compared to the US Pharmacopeia specifications for dissolution of curcuminoid capsules or curcuminoid tablets, the dissolution of the formulated curcumin was not lower than 75% after 60 minutes, suggesting that the formulation greatly increased curcumin’s solubility.

### 3.2. Oral Administration of Formulated Curcumin Shows an Increase in Bioavailability over Conventional Curcumin via Pharmacokinetic Analysis

To determine the actual amount of curcumin that was released and existed in the formulated curcumin, a high-performance liquid chromatography assay and LC-MS method was adopted for the quantification of curcumin. To investigate whether the bioavailability of curcumin was increased after formulation, 60 mg/kg of formulated curcumin was orally administered in a rat model and the plasma samples were subjected to chromatography. To confirm the reliability of the method for analyzing curcumin in plasma samples, a method validation was performed. The retention times of curcumin were about 5.14 minutes, with no visible interference peak in the blank plasma chromatogram (data not shown). To perform a pharmacokinetic analysis, the curcumin concentrations in rat plasma at different time points following oral administration of 500 mg/kg of curcumin and 60 mg/kg of formulated curcumin were compared (Figure 2). However, administration of 500 mg/kg of curcumin had very low amount of curcumin in rat plasma and resulted to a huge increase of bioavailability. Therefore, the pharmacokinetic parameters of curcumin represented an estimated number and will be compared with others from literatures (see later in discussion). AUC represents the total drug exposure over time. Based on the pharmacokinetic parameters (Table 2), the AUC represents the total drug exposure over time. The AUC normalized by dose of curcumin was increased from 0.0021 to 1.864, which is an 887.6-fold increase after formulation. Overall, this result showed that oral administration of formulated curcumin significantly increased the oral bioavailability of curcumin compared with conventional curcumin.

### 3.3. Gene Expression Analysis of Formulated Curcumin and Prediction of Highly Correlated Pathways

We queried CLUE with regard to the analyzed group of genes in order to identify potential biological functions of formulated curcumin. The top 30 compounds (Figure 3C) and PCLs (Figure 3A) with the highest scores were obtained from CLUE. Intersection of results from both cell lines revealed that seven compounds (e.g., menadione and angiogenesis inhibitor; Figure 3D) and two PCLs (e.g., NF-κB pathway inhibitors; Figure 3B) shared common functions with conventional curcumin. In summary, formulated curcumin was similar to conventional curcumin, and both functioned as, for example, NF-κB pathway inhibitors, which is consistent with previous studies [33]. We intersected two sample groups to identify common PCL/compound classes between formulated curcumin treatment of HT29 cells and formulated curcumin treatment of HepG2 cells. Consequently, we identified two PCLs (including NF-κB pathway inhibitors and vesicular transport loss of function (LOF; Figure 3B), and seven compounds belonging to the NF-κB pathway inhibitor PCL class were common among all groups.

In fact, CLUE revealed that only the MOA of compounds or shRNAs was similar to that of formulated curcumin; CLUE did not indicate the target of formulated curcumin. Therefore, to obtain more information about formulated curcumin, we used another database for assistance pathway analysis, ConsensusPathDB (CPDB). CPDB comprises interactions among different types of various intracellular information, such as genes, RNA, proteins, and metabolites, to predict a relatively comprehensive and unbiased cellular biology signal result. We used the Venny website to intersect our two sets of PCL results (Figure 3B) and selected their targets and members genes to predict potential pathways of formulated curcumin (Figure 4). According to *q* < 0.001, we listed the top 20 pathways at the bottom of Figure 4, and the details are in Appendix A. In addition, we showed similar effects in L1000 microarray profiles between conventional and formulated curcumin in heatmaps (Figure 5). These data suggested that formulated curcumin exhibits similar biological functions as conventional curcumin.

### 3.4. Formulated Curcumin Displays Stronger Inhibition on Population Growth of Huh7 Cells

To determine whether formulated curcumin retains the biological functions of conventional curcumin, we performed MTT assay to assess the cell proliferation and cell viability of conventional curcumin and formulated curcumin-treated Huh7 cells. As shown in Appendix A, when the concentration of conventional or formulated curcumin increased, cell viability decreased after treatment with curcumin for 24–120 h. We also observed cytotoxicity of curcumin in Huh7 cells. The 50% inhibitory concentration (IC_50_) value for conventional curcumin at 24, 48, 72, 96, and 120 h was around 90, 90, 60, 45, and 30 μM, respectively, while the IC_50_ value for formulated curcumin was around 60, 60, 30, 10, and 10 μM, respectively.

### 3.5. Cytotoxic Effect of Curcumin on Other HCC Cell Lines

To determine whether conventional and formulated curcumin could mediate the survival of other HCC cell lines, we first examined the effect of conventional and formulated curcumin on the viability of Huh7, Mahlavu, and PLC5 cells using MTT assay. While Huh7 cells are well differentiated, Mahlavu and PLC5 cells are poorly differentiated and carry *p53* mutations. To explore the cytotoxic activity of conventional and formulated curcumin against these HCC cell lines, we initiated an *in vitro* study by treating Huh7, Mahlavu, and PLC5 cells each with increasing dosages of conventional and formulated curcumin (0, 1, 3, 10, 30, 60, and 90 μM) for 72 h. MTT assay results indicated that both conventional and formulated curcumin significantly inhibit the viability of Huh7 (Figure 6A), Mahlavu (Figure 6B), and PLC5 (Figure 6C) cells. After 72 h post-treatment, formulated curcumin caused cytotoxicity in Huh7, Mahlavu, and PLC5 cells with an IC_50_ value of 38.1, 10.1, and 60.9 μM, respectively, while the IC_50_ values of conventional curcumin were 53.3, 35.4, and 82.5 μM, respectively (Figure 6A–C). Taken together, the results showed that both conventional and formulated curcumin inhibited the survival and proliferation of HCC cell lines in a dose-dependent manner. In addition, formulated curcumin was more effective than conventional curcumin.

### 3.6. Effects of Curcumin on Apoptosis-Related Protein Expression

MTT assay indicated that both conventional and formulated curcumin suppress cell viability. To determine whether cell viability inhibition is due to apoptosis, we examined the degree of apoptosis by PI and annexin V staining using flow cytometry. The results clearly demonstrated that 24 h post-treatment with 10–60 μM of conventional and formulated curcumin, the percentage of cells undergoing early apoptosis (annexin V^+^/PI^–^) and late apoptosis (annexin V^+^/PI^+^) increased, while the percentage of viable cells decreased (annexin V^–^/PI^–^) in a dose-dependent manner (Figure 7A). Comparison of effectiveness between conventional and formulated curcumin showed no significant statistical difference.

Caspase-3 activation is crucial for mitochondrial-dependent and mitochondrial-independent apoptotic pathways. Therefore, we examined the activity of caspase-3 by observing its active form (cleaved form) by western blotting in Huh7 cells (Figure 7B). Both conventional and formulated curcumin increased caspase-3 activity in Huh7 cells in a dose-dependent manner (Figure 7B). Caspase-3 activation led to cleavage of several substrates, including PARP. PARP cleavage was also determined by western blotting (Figure 7B). Therefore, conventional and formulated curcumin separately induced apoptosis by activating caspase-3 (Figure 7B).

Many antineoplastic drugs induce apoptosis of cancer cells via mitochondrial apoptotic pathways [34,35,36]. A hallmark of apoptosis induction via these pathways is a rapid, early breakdown of the mitochondrial membrane potential. Therefore, in this study, we also analyzed the mitochondrial function integrity posttreatment with conventional and formulated curcumin. Both conventional and formulated curcumin significantly induced mitochondrial membrane potential breakdown (ΔΨm) in a concentration-dependent manner (Figure 7C), as determined by ELISA using the potential-sensitive dye JC-1.

Previous studies have shown that curcumin inhibits cancer cell proliferation via suppression of the ERK signaling pathway [37,38,39]. To investigate whether the ERK signaling pathway is involved in curcumin-induced apoptosis, ERK activation was evaluated by detecting ERK phosphorylation. Huh7 cells were exposed to 10–60 μM of conventional curcumin and formulated curcumin separately for 48 h, and ERK activation was determined by western blotting. As shown in Figure 7D, both conventional and formulated curcumin-induced phosphor-ERK down-regulation but with little change in the total ERK protein in Huh7 cells.

Recent studies have suggested that a curcumin-induced mitotic spindle defect and cell cycle arrest in human cancer cells occur through Aurora kinase inhibition [40,41,42]. To determine whether conventional and formulated curcumin inhibit Huh7 cell proliferation via down-regulation of Aurora-A expression, Huh7 cells were incubated with 0–60 μM of conventional and formulated curcumin separately for 48 h. We observed a significant decrease in the level of Aurora-A by western blotting (Figure 7D). These data suggested that the biological function of both conventional and formulated curcumin involves caspase-3 activation and ERK and Aurora-A down-regulation.

### 3.7. Effects of Combination of Curcumin and Sorafenib

Sorafenib and curcumin both inhibited cell viability of two HCC cell lines (Huh7 and Hep3B) in a dose-dependent manner (Appendix A). Therefore, we explored the potential effects of sorafenib in combination with conventional or formulated curcumin on Huh7, Mahlavu, and Hep3B cell proliferation. The cells were treated with various concentrations of sorafenib in combination with conventional or formulated curcumin for 48 h (Appendix A). The results indicated that the combination of conventional or formulated curcumin with sorafenib has a stronger inhibitory effect on the population growth of HCC cell lines.

In this study, we demonstrated that both conventional and formulated curcumin inhibit Aurora kinase, preferentially suppress proliferation, and induce apoptosis of HCC cell lines. A previous study also indicated that curcumin induces apoptosis-associated autophagy [43,44]. Therefore, we investigated whether the combination of sorafenib and conventional or formulated curcumin can increase apoptosis of HCC cell lines. Cell cycle analysis revealed that adding conventional curcumin for 48 h leads to an apoptosis rate of 13.5% (sub-G1 population), similar to the 10.9% rate induced by sorafenib. We further examined whether conventional curcumin induces apoptosis of Hep3B cells. We found that the levels of cleaved caspase-3 proteins increased in conventional curcumin-treated, sorafenib-treated, and combination-treated cells (Appendix A). There was no notable difference between the three groups. Conventional curcumin and its combination with sorafenib both significantly induced accumulation of LC3-II (Appendix A), a lipidated form of LC3 that is considered an autophagosomal marker in mammals. Similar results were found with formulated curcumin treatment (data not shown). These results suggested that the combination of conventional curcumin and formulated curcumin separately with sorafenib induces apoptosis-associated autophagy in HCC cell lines.

## 4. Discussion

In this study, we reported that (i) both conventional and formulated curcumin induce apoptosis of HCC cell lines and down-regulate Aurora-A and (ii) a combination of conventional or formulated curcumin with sorafenib has a stronger inhibitory effect on HCC cell viability, demonstrating a possible prevention and therapeutic application for formulated curcumin to be used as a food supplement and medical food.

HCC most often develops and progresses with a lot of oxidative stress and inflammation. Phytochemicals, such as dietary polyphenols, endowed with potent antioxidant and anti-inflammatory properties, provide a suitable alternative for alleviation of HCC. Previous studies have reported that systemic bioavailability of curcumin in humans is very poor [45,46]. Solid dispersion manufacturing of poorly soluble drugs by spray-drying is a practical commercialization strategy to improve solubility and dissolution rates because of the reasonable cost of required materials and ease of scale-up [47].

Several methods have been developed to solve issue of low oral bioavailability of curcumin. One method is the use of a natural enhancer, such as alkaloid piperine. When 20 mg of piperine was given concomitantly with 2 g of curcumin, the serum curcumin increased by 20 times in humans and 1.56 times in rats. However, because piperine is a relatively selective CYP3A inhibitor [14], drug and food interactions with piperine are a concern. Some studies have used phospholipids as a delivery vehicle for curcumin. The oral bioavailability of curcumin-silica-coated flexible liposomes (curcumin-SLs) and curcumin-flexible liposomes (curcumin-FLs) was found to be 7.76- and 2.35-fold higher, respectively, compared to curcumin suspensions [20]. However, large-scale manufacturing will incur a substantial cost.

In the pharmaceutical industry, solid dispersion pellets are one of the drug delivery solutions for poorly soluble drugs. Solid dispersion pellets increase the solubility and release rate of a drug. For example, itraconazole is a triazole antifungal agent with low solubility. In some itraconazole solid dispersion pellets, the formulated itraconazole showed around 30- and 70-fold increase in the dissolution rate compared to pure drug [48]. Tanshinone IIA (TA), one of the liposoluble bioactive constituents extracted from the root of *Salvia miltiorrhiza* Bunge, has positive cardiovascular functions such as vasorelaxation and cardioprotective effects. The oral bioavailability of TA tSD pellets (a solid dispersion of a combination of PVP and poloxamer 188) in rabbit increased by 5.4 times compared to TA [49]. In this study, the oral bioavailability of formulated curcumin (solid dispersion pellets) increased significantly compared to conventional curcumin. Except for curcumin itself, all the materials used are inert and safe. In addition, because of the simple manufacturing process, solid dispersion pellets are also suitable for continuous and large-scale manufacturing, and the formulated curcumin can be used as a food supplement and medical food.

As mentioned in Section 3.1, 830–1000 µm is the size distribution of the pellets (formulated curcumin) determined by the sieving method. These pellets were formed by spraying ethanol solution of curcumin and PVP-K30 onto sugar spheres in a fluid-bed granulator machine. After re-dispersion in water, they were disintegrated to release free curcumin into the water as nanoparticles. The size of the curcumin particles (Z-average) measured using Zetasizer Nano ZS90 was determined to be 141.9 ± 5.1 nm, which could be filtered by a 0.45-µm pore filter. The nano size of the curcumin particles made it possible for them to be dissolved quickly in the dissolution medium. This was compatible with the data of the dissolution test.

In addition, using 1% SDS solution as the media for the dissolution test is mentioned in the US Pharmacopeia for curcuminoid tablets or curcuminoid capsules. Therefore, we used SDS in the medium to evaluate how well our newly formulated curcumin was improved in its dissolution by comparison with conventional curcumin. The results showed that dissolution of conventional curcumin in 1% SDS solution was only about 20% after 60 minutes, whereas that of the formulated curcumin was more than 85% after 10 minutes.

Appendix A summarizes several studies that have reported techniques for enhancing curcumin oral bioavailability. While the formulated curcumin increases bioavailability over 800-fold, only approximately 5–100-fold increases have been achieved by other formulations. To clarify this difference, we investigated several pharmacokinetic parameters of conventional curcumin from these studies (Appendix A) ([50,51,52]). In our study, administration of 500 mg/kg of curcumin resulted in a very low amount of curcumin in rat plasma: AUC_0-t_ (h × ng/mL) was 1.1, whereas in other studies the AUC_0-t_ (h × ng/mL) were 8.76, 80, and 60 after doses of 50, 340, and 500 mg/kg, respectively, were administered. If we used these AUC_0-t_ (h × ng/mL) from un-formulated curcumin to calculate the AUC for the formulated curcumin, an approximately 8–16-fold increase in bioavailability was obtained. In short, this study showed that oral administration of a novel solid dispersion of curcumin significantly increased its oral bioavailability compared with that of conventional curcumin. Although the C_max_ of the formulated curcumin is far below the effective concentrations of our cell culture experiments, administration of conventional curcumin (200 mg/kg) for days or weeks has been reported to exhibit significant biological activity against both chemically induced and xenograft hepatocarcinogenesis [53]. Because the formulated curcumin (single administration of 60 mg/kg) significantly increased the oral bioavailability compared with conventional curcumin (single administration of 500 mg/kg), repeating administration of the formulated curcumin at a higher dosage for a longer period of time can be expected to achieve a much higher C_max_, and especially AUC, to display the biological effect of curcumin observed *in vitro*. It is noteworthy that the formulated curcumin, as expected, retains similar characteristics to conventional curcumin at the cellular level.

It is well known that the low oral bioavailability of curcumin is due to its poor solubility, poor intestinal permeability and extensive metabolism. In another study [52], authors have used phospholipid to formulate liposome of curcumin in order to enhance intestinal absorption of curcumin. In the fasted rat model (rats were fasted overnight to avoid interference by food), they proved that the AUC for liposome of curcumin increased about 5.5-fold compared with that for the conventional curcumin after administration of 340 mg/kg dose; the AUC 26.7 µg × min/mL is equivalent to 445 h × ng/mL. In our study, rats had free access to food and water. After administrating 60 mg/kg of the formulated curcumin, the AUC recorded was approximately 111 h × ng/mL. In addition, AUC/dose calculated in our study was higher than that in the previous study. Although we enhanced solubility of curcumin, we suggest that after administration of our formulation, curcumin was quickly released as nano particles then dissolved in gastric fluid or tiny oil droplet in food. Therefore, a part of curcumin was absorbed as free form and the rest was absorbed by pathway of oil absorption supported by bile salts. This was one of the advantages of the formulated curcumin which can be prepared easily and scaled up in industry. In future, we will conduct more experiments to verify this conclusion.

Both conventional and formulated curcumin were effective in decreasing proliferation and viability of HCC cell lines in a dose-dependent manner and induced apoptosis of HCC cell lines via mitochondria dysfunction *in vitro*. In the cell culture study, both forms of curcumin were dissolved completely before experiments, thus the benefits of formulated curcumin with improved solubility and dissolution rates could not be displayed. The *in vitro* study was carried out mainly to determine if the formulated curcumin retained similar characteristics of conventional curcumin at the cellular level. Mitochondrial hyperpolarization is a prerequisite for curcumin-induced apoptosis, and mitochondrial DNA (mtDNA) damage might be a probable mechanism for curcumin-induced apoptosis of HepG2 cells and might serve as the initial event triggering a chain of events leading to apoptosis [12]. Aurora kinases, such as Aurora-A, Aurora-B, and Aurora-C, comprise a family of centrosome-associated serine/threonine kinases that are overexpressed in various cancers and are potentially correlated with chemoresistance [54,55,56]. Curcumin administration [57] or Aurora-A inhibition by short interfering RNA (siRNA) [58] induces apoptosis. Curcumin has also been shown to down-regulate Notch1, the janus kinase (JAK)/signal transducer and activator of transcription (STAT) pathway, and multidrug resistance protein 1 (MDR1) expression and to inhibit histone deacetylase 1 (HDAC1) activity [12,13,42,59,60]. Sorafenib is the only chemotherapeutic drug that has been shown to be effective in prolonging the survival of HCC patients. However, the low rate of tolerance to sorafenib among HCC patients limits its use [31,32]. Recent preclinical studies have reported that combining sorafenib with other chemotherapeutic agents exerts synergistic effects [61,62], which could provide a promising strategy for the treatment of advanced HCC. In this study, conventional or formulated curcumin in combination with sorafenib inhibited the proliferation of HCC cell lines and exhibited a stronger inhibitory effect on HCC cell lines. Taken together, the formulated curcumin appeared to have properties similar to conventional curcumin, raising the possibility that our formulated curcumin could enhance cytotoxicity against HCC.

In addition, our bioinformatics screening platform effectively identified the molecular mechanisms of a phytochemical via gene expression profiles. C-Map can be queried to identify specific gene signatures from small molecules, including FDA-approved drugs. CLUE is similar to C-Map but considerably larger, with > 1.1 million L1000 profiles; therefore, similarity scores for compounds in CLUE can be obtained to identify their molecular actions. Similar results for conventional and formulated curcumin suggested that only solubility and oral bioavailability have been altered in formulated curcumin. Curcumin is known to exert strong anti-inflammatory effects by interrupting NF-κB signaling at multiple levels. Based on CLUE analysis, formulated curcumin can be predicted to have a similar action as NF-κB pathway inhibitors. Furthermore, analysis via CPDB suggests that curcumin can be linked to TNF related weak inducer of apoptosis as well as TNF mediated NF-κB pathway. Therefore, our screening platform not only confirms that the formulated curcumin has similar mechanism with unformulated curcumin, but it also predicts the novel mechanism of formulated curcumin, such as suppression of HMGB1 mediated inflammation by THBD (label in red in Appendix A).

In conclusion, our curcumin was formulated in pellet form, which not only improved its oral bioavailability but also provided high flexibility of use. For example, it would be easy to adjust the dosage and combine the pellets with other ingredients. In addition, as a functional food and alternative medicine, it would be suitable for those who cannot swallow tablets or capsules.

## Figures and Tables

**Figure 1 nutrients-11-02982-f001:**
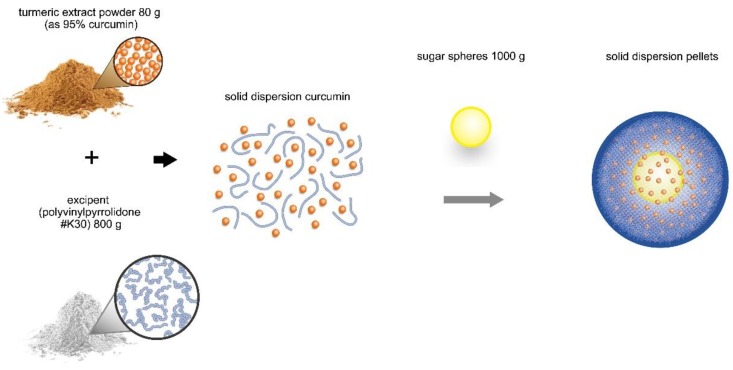
Preparation of solid dispersion curcumin. Turmeric extract powder, containing 80 g of 95% curcumin (as the active drug), was mixed with excipients (800 g PVP as the nonvolatile polymer solvent in the presence of alcohol as the volatile solvent for curcumin). The drug-polymer interaction evenly dispersed curcumin in the solvent. Then, the solvent containing solid dispersion curcumin was loaded onto sugar spheres by fluid-bed granulation to make solid dispersion curcumin pellets. PVP, polyvinylpyrrolidone.

**Figure 2 nutrients-11-02982-f002:**
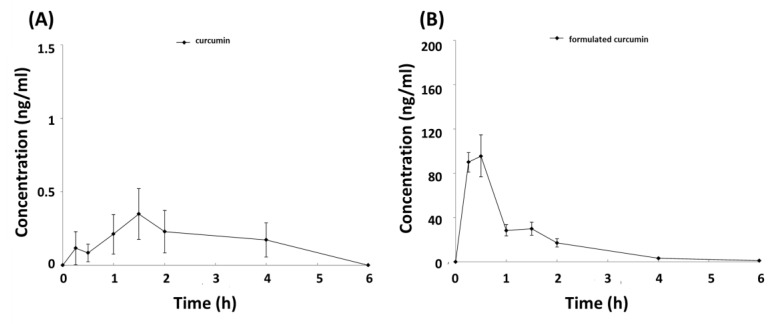
Mean plasma concentration-time profiles of curcumin in male SD rats following 500 mg/kg conventional curcumin (**A**) and 60 mg/ kg formulated curcumin (**B**) after single dose oral gavage (P.O.) linear ordinate. The data are expressed as mean ± SD, *n* = 5 for curcumin and *n* = 6 for formulated curcumin.

**Figure 3 nutrients-11-02982-f003:**
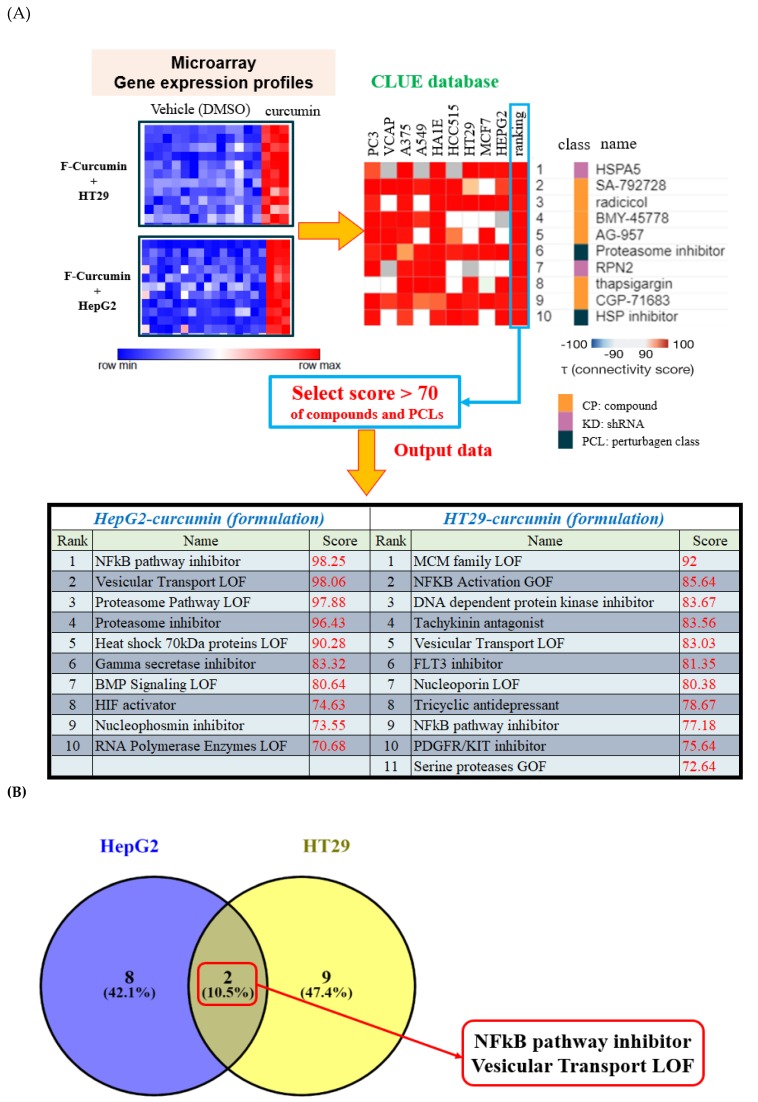
Gene analysis data of formulated curcumin. (**A**) The L1000 gene expression profiles of HT29 and HepG2 cells treated with formulated curcumin were analyzed by CLUE. The output data of PCLs in both HT29 and HepG2 are shown at the bottom (score > 70). (**B**) PCLs list a score > 70 and are intersected by Venny website. Two common PLCs, NF-κB pathway inhibitor and vesicular transport LOF, are shown in the diagram. To avoid missing possible predicted functions of formulated curcumin, we used an intersection-driven approach to broadly cover these PCLs (score > 70). (**C**) The top 30 compounds (CP) are representative, while the complete list is provided in the Appendix A (score > 90). Details are in Appendix A. (**D**) Intersection compounds using L1000 array analysis of formulated curcumin by CLUE. The connectivity score is based on the Kolmogorov–Smirnov enrichment statistical evaluation of each gene expression profile. The results provided from CLUE are expressed as a comprehensive connectivity score, showing that the same drug has a similar MOA on different cancer cells in CLUE, Connectivity Map and Library of Integrated Network-Based Cellular Signatures (LINCS) unified environment; PCL, perturbagen class; MOA, mechanism of action.

**Figure 4 nutrients-11-02982-f004:**
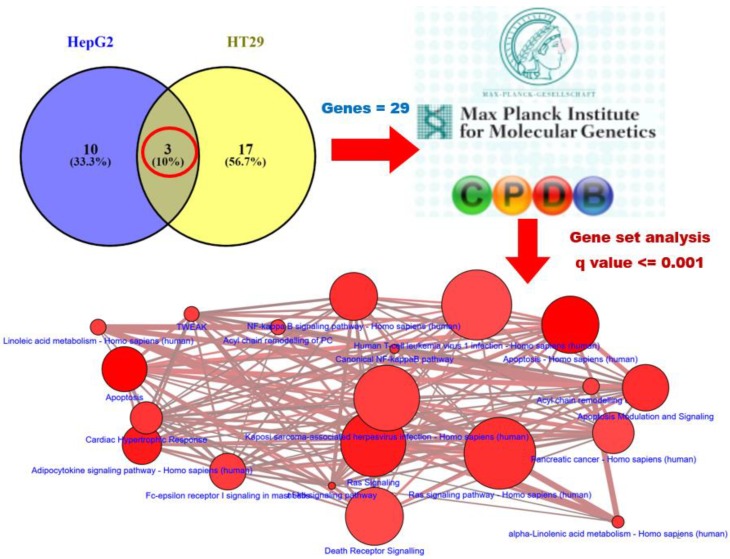
Possible pathways of formulated curcumin predicted by CPDB. Genes in two datasets were used to query CPDB in order to predict the pathways in which these genes were likely participating. The Venn diagram shows the intersecting PCLs (top, right). We focused on intersection results indicated by red circles. The results contained two PCLs, NF-κB pathway inhibitors and vesicular transport LOF and used their targets and member gene lists (total 22 genes) to query CPDB in order to analyze interaction network modules, biochemical pathways, and functional information. Top 50 prediction pathways are listed in Appendix A, and top 20 pathways identified by CPDB analysis (*q* < 0.001) are shown at the bottom of the figure. The size of each dot denotes the entity number of genes in the pathway. The line between two dots was calculated by the function of these two pathways to indicate the number of genes overlapping said pathways. The breadth of the line denotes the strength of the correlation between two dots. The apoptosis was analyzed in this study (highlighted in yellow, Appendix A). CPDB, ConsensusPathDB; PCL, perturbagen class; shRNA, short hairpin RNA.

**Figure 5 nutrients-11-02982-f005:**
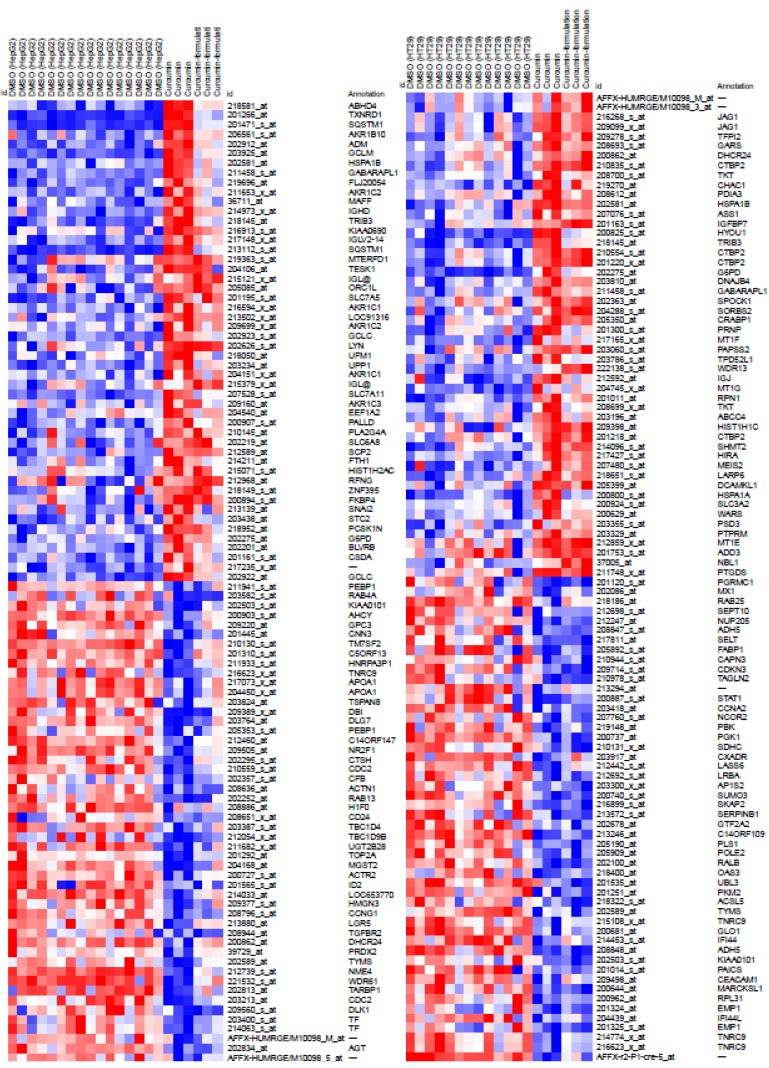
Heatmaps showing the top 50 up/down L1000 probes with similar expression patterns between conventional and formulated curcumin in HepG2 and HT29 cells, respectively. The horizontal axis denotes the treatments in three groups: (1) DMSO, (2) conventional curcumin, and (3) formulated curcumin. The vertical axis denotes the L1000 probe IDs and their corresponding gene names.

**Figure 6 nutrients-11-02982-f006:**
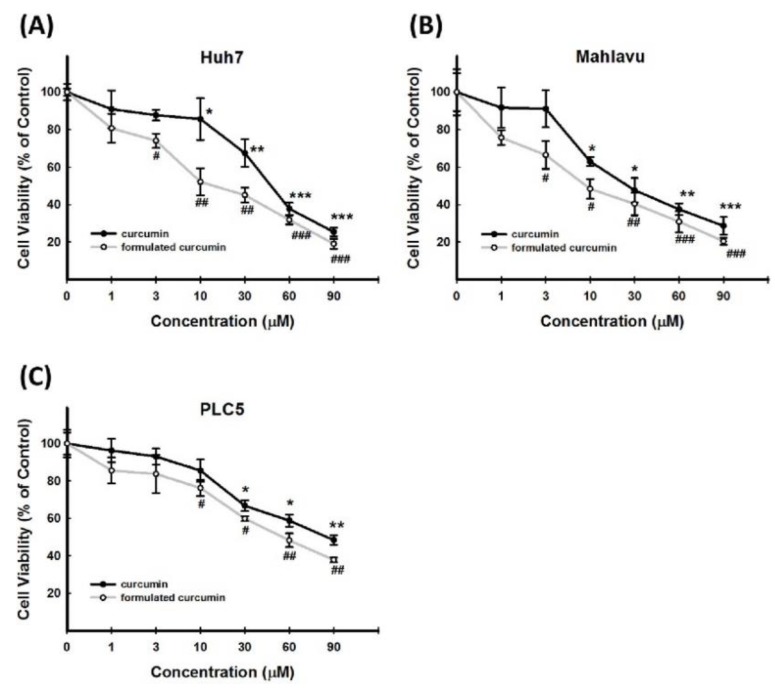
Cytotoxic effect of curcumin and formulated curcumin on other HCC cell lines (Huh7, Mahlavu, and PLC5). After separate treatment with 1, 3, 10, 30, 60, and 90 µM of conventional and formulated curcumin for 72 h, cell viability was determined by MTT assay and expressed as a percentage relative to the control group. Formulated curcumin was more effective than conventional curcumin on the basis of cell viability in the three cell lines tested. Formulated curcumin has higher cytotoxicity in (**A**) Huh7, (**B**) Mahlavu, and (**C**) PLC5 compared to conventional curcumin. ^#^*p* < 0.05; ^##^*p* < 0.01; ^###^*p* < 0.005 compared to the control group (formulated curcumin). **p* < 0.05; ***p* < 0.01; ****p* < 0.005 compared to the control group (conventional curcumin). MTT, 3-(4,5-cimethylthiazol-2-yl)-2,5-diphenyl tetrazolium bromide.

**Figure 7 nutrients-11-02982-f007:**
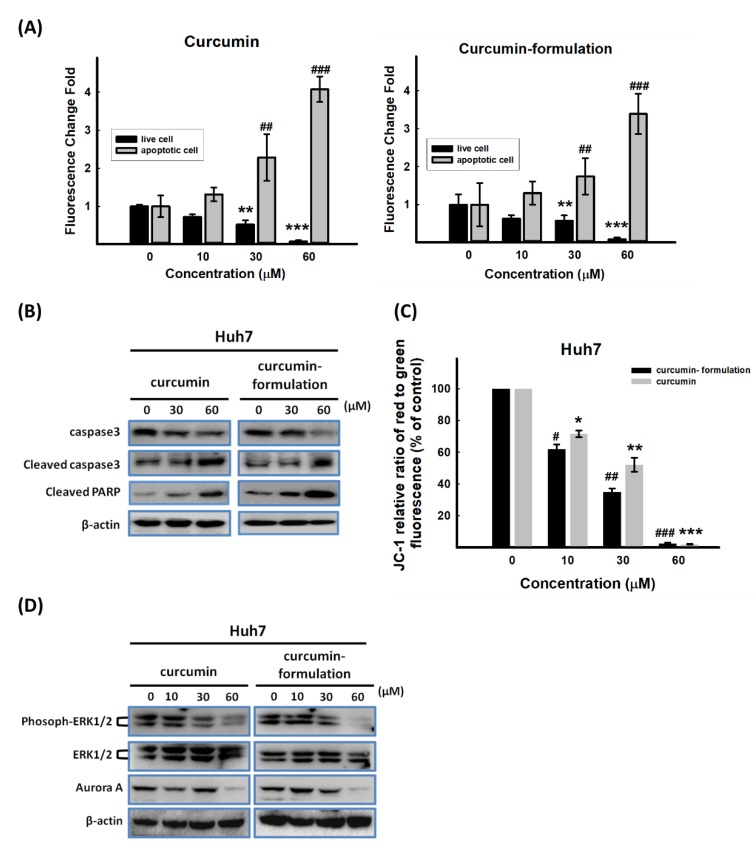
Both conventional and formulated curcumin caused apoptosis of Huh7 cells. (**A**) The cells were incubated and treated with conventional and formulated curcumin. After 24 h, the cells were subjected to annexin V/PI staining and analyzed by flow cytometry. Quantitative analysis of PI- or annexin V-positive cells is shown (*n* = 3). ***p* < 0.01 and ****p* < 0.005 compared to the live-cell group without treatment. ^##^*p* < 0.01 and ^###^*p* < 0.005 compared to the apoptotic cell group without treatment. (**B**) Huh7 cell lysates treated with conventional and formulated curcumin separately were subjected to immunoblot analysis. The expression levels of cleaved caspase-3, caspase-9, and PARP increased, demonstrating that the cells had undergone apoptosis. (**C**) The mitochondrial membrane potential (ΔΨm) in Huh7 cells was analyzed using the JC-1 mitochondrial membrane potential assay. ΔΨm was lower in HCC cell lines treated with different concentrations (10, 30, and 60 µM) of conventional or formulated curcumin compared to control HCC cell lines (n = 3). **p* < 0.05; ***p* < 0.01; and ****p* < 0.005 compared to the group without conventional curcumin treatment. ^#^*p* < 0.05; ^##^*p* < 0.01; and ^###^*p* < 0.005 compared to the group without formulated curcumin treatment. (**D**) P-ERK and AURKA expression was down-regulated by conventional and formulated curcumin in a concentration-dependent manner, as shown by immunoblot analysis results with anti-ERK, anti-P-ERK, and anti-AURKA antibodies. PI, propidium iodide; PARP, poly (ADP-ribose) polymerase; ERK, extracellular regulated protein kinase; AURKA, aurora kinase A.

**Table 1 nutrients-11-02982-t001:** Properties of formulated curcumin.

Properties of Formulated Curcumin	Value
Pellet size (µm)	830–1000
Curcumin level (%)	3.3
Z-average (nm)	141.9 ± 5.1
Polydispersity index (PDI)	0.308 ± 0.029
Zeta potential (mV)	–2.49 ± 0.39

**Table 2 nutrients-11-02982-t002:** Pharmacokinetic parameters of curcumin in rat plasma following oral administration.

Parameters	Oral	
	Conventional Curcumin* (*n* = 5)	Formulated Curcumin (*n* = 6)
	500 mg/kg	60 mg/kg
C_max_ (ng/mL)	0.704 ± 0.272	109.200 ± 41.651
AUC_0-t_ (h × ng/mL)	1.1 ± 1.2	111.8 ± 16.4
AUC_0-t_/Dose	0.0022 ± 0.0024	1.863 ± 0.273
T_max_(h)	1.25 ± 0.83	0.38 ± 0.14

The data are expressed as the mean ± SD. C_max_: the maximum plasma concentration; T_max_: the time at which *C_max_* is observed; AUC_0-t_: area under the concentration-time curve from the time of drug administration to the last quantifiable concentration. *: Conventional curcumin in some rat plasma samples was very low and was assigned to 0 for calculations. We were unable to detect curcumin from the plasma of one rat during the experimental periods, and thus data from five rats were used for calculations.

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
