# Peer review of "Bioactivity Evaluation of a Novel Formulated Curcumin"

_nutrients, 2019, doi:10.3390/nu11122982_

Round 1

Reviewer 1 Report

The paper by Liao et al investigated a new curcumin formulation in rats and liver cell lines. They present data to suggest improved bioavailability and comparable cell culture effects of the new pellet form of curcumin compared to unformulated reagent-grade curcumin, an intriguing finding. Although the authors did not demonstrate that the new form of curcumin is more biologically potent than curcumin itself, they seem to be headed in that direction.  The paper is well-written in that minimal editing for clarity and grammar is required, although some additional methodological content is needed as noted.

Specific comments:

The abstract should be revised to more comprehensively include the gene profiling experiments performed and the results obtained from those experiments. The main finding, that the new curcumin formulation has the same effects on gene expression as regular curcumin, should be stated more explicitely.

The curcumin assay used for Fig 1 should be described. Was curcumin (as opposed to a metabolite) specifically identified? Figure 1 suggests a peak plasma concentration of 0.3 microM, while higher concentrations were required for cell culture effects (>10microM for significance in MTT, and >30 microM for apoptotic effects). This deserves comment.

Figure 7A is confusing. It is not clear how the experiment was done or what is being plotted. Figure 7C is also confusing but may be fixed by relabeling the graph. Exactly what is “control” if the 0 concentration point has error bars (indicating s?)

The description of methods used to distinguish additive from synergistic effects is inadequate. I do not believe the equation given in the methods is correct and I could not find the reference offered in its support (“Jin’s method”). This is a minor part of the paper and should be omitted without a more rigorous treatment. Similarly, given that there does not seem to be any particular relationship between sorafenib and curcumin, Figure 8 should be omitted.

In summary, documenting the improved bioavailability in rats of a new form of curcumin that retains curcumin’s concentration dependence of cell culture apoptotic effects of is an important result.

Author Response

Response to Reviewer comments for nutrients-624955

Title: Bioactivity evaluation of a novel formulated curcumin

Author(s): Se-Chun Liao, Wei-Hsiang Hsu, Zi-Yi Huang, Kun-Lin Chuang, Kuan-Ting Lin, Chia-Ling Tseng, Tung-Hu Tsai, Anh-Hoang Dao, Chun-Li Su *, Chi-Ying F. Huang *

Dear Editor:

We would like to thank the reviewers for their thorough reading of our manuscript as well as their valuable comments. We have conscientiously re-examined the manuscript and made certain revisions. Enclosed here with are the itemized changes. We have followed their comments closely and feel that their suggestions have further strengthened the manuscript. Below are our point-by-point responses.

Reviewer comments:

Reviewer #1 (Remarks to the Author):

The paper by Liao et al investigated a new curcumin formulation in rats and liver cell lines. They present data to suggest improved bioavailability and comparable cell culture effects of the new pellet form of curcumin compared to unformulated reagent-grade curcumin, an intriguing finding. Although the authors did not demonstrate that the new form of curcumin is more biologically potent than curcumin itself, they seem to be headed in that direction.  The paper is well-written in that minimal editing for clarity and grammar is required, although some additional methodological content is needed as noted.

Comment 1. The abstract should be revised to more comprehensively include the gene profiling experiments performed and the results obtained from those experiments. The main finding, that the new curcumin formulation has the same effects on gene expression as regular curcumin, should be stated more explicitely.

Response: As suggested, the abstract has been revised, and more explicitly descriptions have been added (lines 27-36).

Comment 2. The curcumin assay used for Fig 1 should be described. Was curcumin (as opposed to a metabolite) specifically identified? Figure 1 suggests a peak plasma concentration of 0.3 microM, while higher concentrations were required for cell culture effects (>10 microM for significance in MTT, and >30 microM for apoptotic effects). This deserves comment.

Response: As suggested, the curcumin assay has been added in the Materials and Methods (section 2.2.3, lines 119-128). In addition, the descriptions have been added in the Discussion: “Although the Cmax of our formulated curcumin is far below the effective concentrations of our cell culture experiments, administration of conventional curcumin (200 mg/kg) for days or weeks has been reported to exhibit significant biological activity against chemical-induced hepatocarcinogenesis and xenograft growth of hepatocarcinogenesis [1]. Since our formulated curcumin (single administration of 60 mg/kg) significantly increased the oral bioavailability compared to conventional curcumin (single administration of 500 mg/kg), repeating administration a higher dosage of our formulated curcumin for a longer period of time can be expected to achieve a much higher Cmax, and especially AUC, to display the biological effect of curcumin observed in vitro. It is noteworthy that, our formulated curcumin, as we desired, retains similar characteristics to conventional curcumin at the cellular level.” (lines 273-275)

Comment 3. Figure 7A is confusing. It is not clear how the experiment was done or what is being plotted. Figure 7C is also confusing but may be fixed by relabeling the graph. Exactly what is “control” if the 0 concentration point has error bars (indicating s?)

Response: Thanks for this valuable comment. Figure 7A showed the degree of apoptosis by PI and annexin V staining using flow cytometry. The live cells and the cells underwent apoptosis at the early and late stages were determined. To make it clear, the descriptions has been changed to “---, the percentage of cells undergoing early apoptosis (annexin V+/PI) and late apoptosis (annexin V+/PI+) increased, while the percentage of viable cells decreased (annexin V/PI) ---“. The methods of PI and annexin V staining using flow cytometry have been added in the Materials and Methods (section 2.7, page 5, lines 201-208). In addition, we agree with that the control value should be zero. Therefore, we re-analyzed the statistics and re-plotted the Figure 7C.

Comment 4. The description of methods used to distinguish additive from synergistic effects is inadequate. I do not believe the equation given in the methods is correct and I could not find the reference offered in its support (“Jin’s method”). This is a minor part of the paper and should be omitted without a more rigorous treatment. Similarly, given that there does not seem to be any particular relationship between sorafenib and curcumin, Figure 8 should be omitted.

Response: Thanks for your comments. As suggested, the descriptions regarding combination effects of drug interactions have been removed, and Figure 8 has been moved to supplementary Figure 2.

References

Darvesh, A. S.; Aggarwal, B. B.; Bishayee, A., Curcumin and liver cancer: a review. Current pharmaceutical biotechnology 2012, 13, (1), 218-28.

Reviewer 2 Report

In this manuscript, a new curcumin formulation was developed and tested by in vivo and in vitro studies.

The findings are interesting, but the overall study design and the draft of this study need to be improved.

The major innovation is the new formulation. However, no information about the new formulation was presented, except for the preparation process. Because all the other studies were based on the new formulation, the new formulation needs to be evaluated, such as particle size, zeta potential, dissolution, etc.

Although the PK study showed the bioavailability increased significantly, the Cmax is still very low (~100 ng/ml, which is around 36 nM which is far below the effective concentration as shown in the in vitro studies (the IC50s were estimated to be greater than 10 uM. Therefore, such increases may not be meaningful to be active in vivo. In addition, the authors concluded that the formulated curcumin had similar characteristics to conventional curcumin at the cellular level. That means the new formulation is not superior to the conventional formulation, so what is the rationale for developing such formulation.

The other concerns are listed below:

Uncompleted sentences such as line 12: code??; line 110: city?? A careful proofread and English edition are urged. “enzyme-linked immunosorbent assay reader” should be a regular plate reader. The structure needs to be reformatted. Some sections can be combined to make it easier for readers, e.g., sections 2.2, 2.3, 2.4, 2.5, and 2.6 should be put under a single title “pharmacokinetics”; Section 2.7 to 2.14 should also be restructured. Where are the samples collected for Western Blot (2.11)? For the cytotoxicity study, IC50 should be calculated instead of comparing the cell viability directly.

Author Response

Response to Reviewer comments for nutrients-624955

Title: Bioactivity evaluation of a novel formulated curcumin

Author(s): Se-Chun Liao, Wei-Hsiang Hsu, Zi-Yi Huang, Kun-Lin Chuang, Kuan-Ting Lin, Chia-Ling Tseng, Tung-Hu Tsai, Anh-Hoang Dao, Chun-Li Su *, Chi-Ying F. Huang *

Dear Editor:

We would like to thank the reviewers for their thorough reading of our manuscript as well as their valuable comments. We have conscientiously re-examined the manuscript and made certain revisions. Enclosed here with are the itemized changes. We have followed their comments closely and feel that their suggestions have further strengthened the manuscript. Below are our point-by-point responses.

Reviewer comments:

Reviewer #2 (Remarks to the Author):

In this manuscript, a new curcumin formulation was developed and tested by in vivo and in vitro studies. The findings are interesting, but the overall study design and the draft of this study need to be improved.

Comment 1. The major innovation is the new formulation. However, no information about the new formulation was presented, except for the preparation process. Because all the other studies were based on the new formulation, the new formulation needs to be evaluated, such as particle size, zeta potential, dissolution, etc.

Response: As suggested, we have performed additional experiments and the new data are in Table 1. The particle size of formulated curcumin (pellets) estimated by sieving is distributed in the range from 830 to 1000 μm. The curcumin content was 3.3% quantified by HPLC. Particle size (Z-average, nm) and polydispersity index (PDI) and ζ-potential (ZP, mV) of curcumin particles after re-dispersing in water were 141.9 ±5.1, 0.308±0.029 and -2.49±0.39, respectively. Dissolution test was conducted by USP 41 paddle method (apparatus 2). The dissolution rates of conventional curcumin were 0%, 10.88%, 14.07%, 19.00%, 20.37%, and 23.35%, while formulated curcumin were 0%, 86.23%, 87.18, 86.8%, and 87.1% at 0 min, 10 min, 20 min, 30 min, 45 min, and 60 min, respectively. In the 1% sodium lauryl sulfate medium, ~85% formulated curcumin was almost immediately released from pellets after 10 minutes. Additionally, if comparing to specification dissolution of curcuminoid capsule or tablet in USP, dissolution of conventional curcumin was not lower than 75% after 60 minutes, suggesting that our formulation had an increase of curcumin’s solubility.

Comment 2. Although the PK study showed the bioavailability increased significantly, the Cmax is still very low (~100 ng/ml, which is around 36 nM which is far below the effective concentration as shown in the in vitro studies (the IC50s were estimated to be greater than 10 uM. Therefore, such increases may not be meaningful to be active in vivo. In addition, the authors concluded that the formulated curcumin had similar characteristics to conventional curcumin at the cellular level. That means the new formulation is not superior to the conventional formulation, so what is the rationale for developing such formulation.

Response: We agree with this valuable comment. The goal of this study is to develop a new formulation of curcumin, which exhibits higher oral bioavailability and remains the characteristics of conventional curcumin. In the cell culture study, both forms of curcumin were dissolved completely before experiments, thus the benefits of formulated curcumin with improved solubility and dissolution rates could not be compared in vitro. Therefore, the following descriptions have been added in the Abstract and Discussion, respectively.

Abstract (lines 21-35)

 “Therefore, the formulated curcumin increases the oral bioavailability in animals, and, as we desired, retains similar characteristics to conventional curcumin at the cellular level.”

Discussion

“In the cell culture study, both forms of curcumin were dissolved completely before experiments, thus the benefits of formulated curcumin with improved solubility and dissolution rates could not be displayed. The in vitro study was carried out mainly to determine if the formulated curcumin retained similar characteristics of conventional curcumin at the cellular level.”

In addition, the following descriptions have been added in the Discussion to explain the potential effectiveness of our formulated curcumin in animals: “Although the Cmax of our formulated curcumin is far below the effective concentrations of our cell culture experiments, administration of conventional curcumin (200 mg/kg) for days or weeks has been reported to exhibit significant biological activity against chemical-induced hepatocarcinogenesis and xenograft growth of hepatocarcinogenesis [1]. Since our formulated curcumin (single administration of 60 mg/kg) significantly increased the oral bioavailability compared to conventional curcumin (single administration of 500 mg/kg), repeating administration a higher dosage of our formulated curcumin for a longer period of time can be expected to achieve a much higher Cmax, and especially AUC, to display the biological effect of curcumin observed in vitro. It is noteworthy that, our formulated curcumin, as we desired, retains similar characteristics to conventional curcumin at the cellular level.” (lines 273-275)

Comment 3. Uncompleted sentences such as line 12: code??; line 110: city?? A careful proofread and English edition are urged. “enzyme-linked immunosorbent assay reader” should be a regular plate reader. The structure needs to be reformatted. Some sections can be combined to make it easier for readers, e.g., sections 2.2, 2.3, 2.4, 2.5, and 2.6 should be put under a single title “pharmacokinetics”; Section 2.7 to 2.14 should also be restructured. Where are the samples collected for Western Blot (2.11)? For the cytotoxicity study, IC50 should be calculated instead of comparing the cell viability directly.

Response: Thanks for your suggestion. The uncompleted sentences have been fixed. The manuscript has been carefully proofread and English edited. As suggested, the name of the plate reader has been corrected. The structure in the Materials and Methods has been reformatted. Samples collection for Western Blot has been added in the Materials and Methods (section 2.8; line 210-211). The IC50 has been calculated, and the following descriptions have been added in the Results: “For the cytotoxicity study, after 72 h post-treatment, formulated curcumin caused cytotoxicity in Huh7, Mahlavu, and PLC5 cells with an IC50 value of 38.1, 10.1, and 60.9 μM, respectively, while the IC50 values of conventional curcumin were 53.3, 35.4, and 82.5 μM, respectively.” (lines 313-316)

References

Darvesh, A. S.; Aggarwal, B. B.; Bishayee, A., Curcumin and liver cancer: a review. Current pharmaceutical biotechnology 2012, 13, (1), 218-28.

Round 2

Reviewer 2 Report

In the revised version, the authors have added more experimental data and improved the quality of the draft. However, the new results brought up another concern about the formulation. The particle size was determined to be 830 to 1000 “µm” (assumed to be nm, section 3.1). If these were true, the dissolution data would be confusing. Because the dissolution samples were filtered by a 0.45 µm filter, the concentration of curcumin should be “free” species (not incorporated in the microsphere). The dissolution results showed that more than 85% of free curcumin was released in 10 minutes (another minor concern here: if the dissolution completed in 10 min, more data points should be presented before 10 min to show the entire curve). Therefore, it is expected that curcumin would be released from the formulation rapidly in vivo. At this stage, the formulation is no longer effective in terms of enhancing drug absorption. Although it can claim that new formulation increased the solubility, such a little increase in solubility would not be significantly meaningful.

In addition, if the presence of SDS may interact with the formulation, it should be excluded in the media.

Therefore, I still have a concern about the significance of the study.

By the way, the text with track changing makes it hard to read especially for the tables. Please modify. Thanks!

Author Response

Response to Reviewer comments for nutrients-624955

Title: Bioactivity evaluation of a novel formulated curcumin

Author(s): Se-Chun Liao, Wei-Hsiang Hsu, Zi-Yi Huang, Kun-Lin Chuang, Kuan-Ting Lin, Chia-Ling Tseng, Tung-Hu Tsai, Anh-Hoang Dao, Chun-Li Su *, Chi-Ying F. Huang *

Dear Editor:

We would like to thank the reviewers for their thorough reading of our manuscript as well as their valuable comments. We have conscientiously re-examined the manuscript and made certain revisions. Enclosed herewith are the itemized changes. We have followed their comments closely and feel that their suggestions have further strengthened the manuscript. Below are our point-by-point responses.

Reviewer comments:

Reviewer #2 (Remarks to the Author):

Comment 1. In the revised version, the authors have added more experimental data and improved the quality of the draft. However, the new results brought up another concern about the formulation. The particle size was determined to be 830 to 1000 “µm” (assumed to be nm, section 3.1). If these were true, the dissolution data would be confusing. Because the dissolution samples were filtered by a 0.45 µm filter, the concentration of curcumin should be “free” species (not incorporated in the microsphere). The dissolution results showed that more than 85% of free curcumin was released in 10 minutes (another minor concern here: if the dissolution completed in 10 min, more data points should be presented before 10 min to show the entire curve). Therefore, it is expected that curcumin would be released from the formulation rapidly in vivo. At this stage, the formulation is no longer effective in terms of enhancing drug absorption. Although it can claim that new formulation increased the solubility, such a little increase in solubility would not be significantly meaningful.

In addition, if the presence of SDS may interact with the formulation, it should be excluded in the media.

Therefore, I still have a concern about the significance of the study.

By the way, the text with track changing makes it hard to read especially for the tables. Please modify. Thanks!

Response: Thanks for this valuable comment. We have made the following changes in the Discussion section and performed additional experiments as following:

As mentioned in section 3.1, 830-1000 µm (µm is correct) is the size distribution of the pellets (formulated curcumin) determined by the sieving method. These pellets were formed by spraying ethanol solution of curcumin and PVP-K30 onto sugar spheres in a fluid-bed granulator machine. After re-dispersion in water, they were disintegrated to release free curcumin into the water as nanoparticles. The size of the curcumin particles (Z-average) measured using Zetasizer Nano ZS90 was determined to be 9±5.1 nm, which could be filtered by a 0.45 µm pore filter. The nano size of the curcumin particles made it possible for them to be dissolved quickly in the dissolution medium. This was compatible with the data of the dissolution test. (line 429-436)

Your concern about using 1% SDS solution as media is highly appreciated. We have made the following changes to make this clearer: “Using 1% SDS solution as the media for the dissolution test is mentioned in the US Pharmacopeia for curcuminoid tablets or curcuminoid capsules. Therefore, we used SDS in the medium to evaluate how well our newly formulated curcumin was improved in its dissolution by comparison with conventional curcumin. The results showed that dissolution of conventional curcumin in 1% SDS solution was only about 20% after 60 minutes, whereas that of the formulated curcumin was more than 85% after 10 minutes.” (line 437-442)

We have performed an additional experiment to add more data points to show the entire curve. The data are included in the results (Fig. S6) (line 249-261).

It is well known that the low oral bioavailability of curcumin is due to its poor solubility, poor intestinal permeability and extensive metabolism. In another study [1], authors have used phospholipid to formulate liposome of curcumin in order to enhance intestinal absorption of curcumin. In the fasted rat model (rats were fasted overnight to avoid interference by food, they proved that the AUC for liposome of curcumin increased about 5.5-fold compared with that for the conventional curcumin after administration of 340 mg/kg dose; the AUC 26.7 µg*min/ml is equivalent to 445 ng*h/ml. In our study, rats had free access to food and water. After administrating 60 mg/kg of the formulated curcumin, the AUC recorded was approximately 111 ng*h/ml. In addition, AUC/dose calculated in our study was higher than that in the previous study. Although we enhanced solubility of curcumin, we suggest that after administration of our formulation, curcumin was quickly released as nano particles then dissolved in gastric fluid or tiny oil droplet in food. Therefore, a part of curcumin was absorbed as free form and the rest was absorbed by pathway of oil absorption supported by bile salts. This was one of the advantages of the formulated curcumin which can be prepared easily and scaled up in industry. In future, we will conduct more experiments to verify this conclusion. (line 464-478)

As suggested, we have provided revised versions with and without tracking.

Reference

Marczylo, T. H.; Verschoyle, R. D.; Cooke, D. N.; Morazzoni, P.; Steward, W. P.; Gescher, A. J., Comparison of systemic availability of curcumin with that of curcumin formulated with phosphatidylcholine. Cancer Chemother Pharmacol 2007, 60, (2), 171-7.
